# Coinfections by noninteracting pathogens are not independent and require new tests of interaction

**Frédéric M. Hamelin**[1], **Linda J. S. Allen**[2], **Vrushali A. Bokil**[3], **Louis J. Gross**[4], **Frank M. Hilker**[5], **Michael J. Jeger**[6], **Carrie A. Manore**[7], **Alison G. Power**[8], **Megan A. Rúa**[9], **Nik J. Cunniffe**[10]*

1 IGEPP, Agrocampus Ouest, INRA, Université de Rennes 1, Université Bretagne-Loire, Rennes, France, 2 Department of Mathematics and Statistics, Texas Tech University, Lubbock, Texas, United States of America, 3 Department of Mathematics, Oregon State University, Corvallis, Oregon, United States of America, 4 National Institute for Mathematical and Biological Synthesis, University of Tennessee, Knoxville, Tennessee, United States of America, 5 Institute of Environmental Systems Research, School of Mathematics and Computer Science, Osnabrück University, Osnabrück, Germany, 6 Centre for Environmental Policy, Imperial College London, Ascot, United Kingdom, 7 Theoretical Biology and Biophysics, Los Alamos National Laboratory, Los Alamos, New Mexico, United States of America, 8 Department of Ecology and Evolutionary Biology, Cornell University, Ithaca, New York, United States of America, 9 Department of Biological Sciences, Wright State University, Dayton, Ohio, United States of America, 10 Department of Plant Sciences, University of Cambridge, Cambridge, United Kingdom

* njc1001@cam.ac.uk

**Data Availability Statement:** All relevant data are within the paper and its Supporting Information files.

## Abstract

If pathogen species, strains, or clones do not interact, intuition suggests the proportion of coinfected hosts should be the product of the individual prevalences. Independence consequently underpins the wide range of methods for detecting pathogen interactions from cross-sectional survey data. However, the very simplest of epidemiological models challenge the underlying assumption of statistical independence. Even if pathogens do not interact, death of coinfected hosts causes net prevalences of individual pathogens to decrease simultaneously. The induced positive correlation between prevalences means the proportion of coinfected hosts is expected to be higher than multiplication would suggest. By modelling the dynamics of multiple noninteracting pathogens causing chronic infections, we develop a pair of novel tests of interaction that properly account for nonindependence between pathogens causing lifelong infection. Our tests allow us to reinterpret data from previous studies including pathogens of humans, plants, and animals. Our work demonstrates how methods to identify interactions between pathogens can be updated using simple epidemic models.

## Introduction

It is increasingly recognised that infections often involve multiple pathogen species or strains/clones of the same species [1, 2]. Infection by one pathogen can affect susceptibility to subsequent infection by others [3, 4]. Coinfection can also affect the severity and/or duration of

**Funding:** This work was initiated during the Multiscale Vectored Plant Viruses Working Group at the National Institute for Mathematical and Biological Synthesis, supported by the National Science Foundation through NSF Award ₩DBI-1300426 (LJG), with additional support from The University of Tennessee, Knoxville (LJG). This material is based upon research supported by the Thomas Jefferson Fund of the Embassy of France in the United States (FMH and VAB), the FACE Foundation (FMH and VAB), and Girton College, University of Cambridge (NJC). The funders had no role in study design, data collection and analysis, decision to publish, or preparation of the manuscript.

**Competing interests:** The authors have declared that no competing interests exist.

**Abbreviations:** AIC, Akaike information criterion; HPV, human papillomavirus; NiDP, Noninteracting Distinct Pathogens; NiSP, Noninteracting Similar Pathogens; S-I-S, susceptible-infected-susceptible.

infection, as well as the extent of symptoms and the level of infectiousness [5]. Antagonistic, neutral, and facilitative interactions are possible [6, 7]. Coinfection therefore potentially has significant epidemiological, clinical, and evolutionary implications [8–10].

However, detecting and quantifying biological interactions between pathogens is notoriously challenging [11, 12]. In pathogens of some host taxa, most notably plant pathogens, biological interactions can be quantified by direct experimentation [13]. However, often ethical considerations mean this is impossible, and so any signal of interaction must be extracted from population-scale data. Analysis of longitudinal data remains the gold standard [14], although the associated methods are not infallible [15]. However, collecting longitudinal data requires a dedicated and intensive sampling campaign, meaning in practice cross-sectional data are often all that are available. Methods for cross-sectional data typically concentrate on identifying deviation from statistical independence, using standard methods such as $\chi^2$ tests or log-linear modelling to test whether the observed probability of coinfection differs from the product of the prevalences of the individual pathogens [16–26]. Detecting such a nonrandom statistical association between pathogens is then taken to signal a biological interaction. The underlying mechanism can range, for example, from individual-scale direct effects on within-host pathogen dynamics [13, 27], to indirect within-host immune-mediated interactions [28], to indirect population-scale 'ecological interference' caused by competition for susceptible hosts [29, 30].

A well-known difficulty is that factors other than biological interactions between pathogens can drive statistical associations. For instance, host heterogeneity—that some hosts are simply more likely than others to become infected—can generate positive statistical associations, since coinfection is more common in the most vulnerable hosts. Heterogeneity in host age can also generate statistical associations, as infections accumulate in older individuals [31–33]. Methods aimed at disentangling such confounding factors have been developed but show mixed results in detecting biological interactions [34–37]. Methods using dynamic epidemiological models to track coinfections are also emerging, although more often than not requiring longitudinal data [38–42].

More fundamentally, however, the underpinning and long-standing assumption that noninteraction implies statistical independence [43, 44] has not been challenged. Here, we confront the intuition that biological interactions can be detected via statistical associations, demonstrating how simple epidemiological models can change the way we think about biological interactions. In particular, we show that noninteracting pathogens should not be expected to have prevalences that are statistically independent. Coinfection by noninteracting pathogens is more probable than multiplication would suggest, invalidating any test invoking statistical independence.

The paper is organised as follows. First, we use a simple epidemiological model to show that the probability that a host is coinfected by both of a pair of noninteracting pathogens is greater than the product of the net prevalences of the individual pathogens. Second, we extend this result to an arbitrary number of noninteracting pathogens. This allows us to construct a novel test for biological interaction, based on testing the extent to which coinfection data can be explained by our epidemiological models in which pathogens do not interact. Different versions of this test, conditioned on the form of available data and whether coinfections are caused by different pathogen species, allow us to reinterpret a number of previous reports [17, 22, 45–51]. Our examples include plant, animal, and human pathogens, and the methodology can potentially be applied to any cross-sectional survey data tracking coinfection.

## Results

### Two noninteracting pathogens

**Dynamics of the individual pathogens.** We consider two distinct pathogen species, strains, or clones (henceforth 'pathogens'), which we assume do not interact; i.e., the

interaction between the host and one of the pathogens is entirely unaffected by its infection status with respect to the other. Epidemiological properties that are therefore unaffected by the presence or absence of the other pathogen include initial susceptibility; within-host dynamics, including rates of accumulation and/or movement within tissues; host responses to infection; and onward transmission. Assuming a fixed-size host population and susceptible-infected-susceptible (S-I-S) dynamics [52], the proportion of the host population infected by pathogen $i \in \{1,2\}$ follows

$$\dot{I}_i = \beta_i I_i (1 - I_i) - \mu I_i, \tag{1}$$

in which the dot denotes differentiation with respect to time, $\beta_i$ is a pathogen-specific infection rate, and $\mu$ is the host's natural death rate.

Whereas natural mortality may be negligible for acute infections, it cannot be neglected for chronic (i.e., long-lasting) infections, which are responsible for a large fraction of coinfections in humans and animals [3, 53]. Likewise, plants remain infected over their entire lifetime following infection by most pathogens, including almost all plant viruses, as well as the anther smut fungus, which drives one of our examples here [45].

We assume that the disease-induced death rate (virulence) is zero, as otherwise there would be ecological interactions between pathogens [30]. However, our model can be extended to handle pathogen-specific rates of clearance (S1 Text, Section 4; S2 Text, Section 3).

**Tracking coinfection.** Making identical assumptions, but instead distinguishing hosts infected by different combinations of pathogens, leads to an alternate representation of the dynamics. We denote the proportion of hosts infected by only one of the two pathogens by $J_i$, with $J_{1,2}$ representing the proportion coinfected. Pathogen-specific net forces of infection are

$$F_i = \beta_i I_i = \beta_i (J_i + J_{1,2}), \tag{2}$$

and so

$$\begin{aligned}
\dot{J}_1 &= F_1 J_\varnothing - (F_2 + \mu) J_1, \\
\dot{J}_2 &= F_2 J_\varnothing - (F_1 + \mu) J_2, \\
\dot{J}_{1,2} &= F_2 J_1 + F_1 J_2 - \mu J_{1,2},
\end{aligned} \tag{3}$$

in which $J_\varnothing = 1 - J_1 - J_2 - J_{1,2}$ is the proportion of hosts uninfected by either pathogen (Fig 1).

**Prevalence of coinfected hosts.** We assume the basic reproduction number, $R_{0,i} = \beta_i/\mu > 1$, for both pathogens. Solving Eq 3 numerically for arbitrary but representative parameters (Fig 2A) shows the proportion of coinfected hosts ($J_{1,2}$) to be larger than the product of the individual prevalences ($P = I_1 I_2$ from Eq 1). That $J_{1,2}(t) \geq P(t)$ for large $t$ (for all parameters) can be proved analytically (S1 Text, Section 1.1). Numerical exploration of the model suggests that $J_{1,2}(t)$ invariably becomes larger than $P(t)$ relatively rapidly, and well within the lifetime of an average host, over a wide range of initial conditions and plausible sets of parameter values (S1 Text, Section 1.2; S1 Fig).

Simulations of a stochastic analogue of the model (Fig 2B) reveal the key driver of this behaviour. The net prevalences of the pathogens considered in isolation, $I_1$ and $I_2$, are positively correlated (Fig 2C; Eq 27 in Methods section 'Stochastic models'), because of simultaneous reductions whenever coinfected hosts die. The full distribution of point estimates of the relative deviation from statistical independence (see Eq 5) indicates the deviation is reliably greater than zero across an ensemble of runs of our stochastic model (Fig 2D). That the deviation is routinely positive is robust to alternative formulations of the stochastic model including environmental as well as demographic noise (S1 Text, Section 6.1; S2 Fig). It also becomes

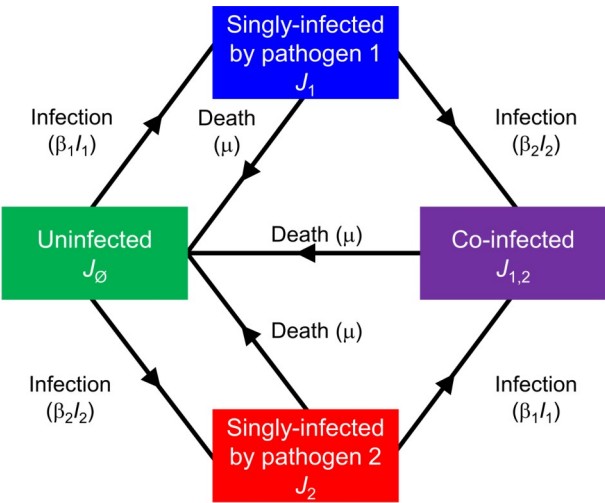

**Fig 1. Schematic of the model tracking a pair of noninteracting pathogens.** The model is defined in Eqs 1–3: $J_\varnothing$ denotes uninfected hosts; $J_1$ and $J_2$ are hosts singly infected by pathogens 1 and 2, respectively; $J_{1,2}$ are coinfected hosts; $I_1 = J_1 + J_{1,2}$ and $I_2 = J_2 + J_{1,2}$ are net densities of hosts infected by pathogens 1 and 2, respectively.

apparent quickly across a wide range of initial conditions; i.e., the sign and magnitude of the relative deviation are relatively robust to transient behaviour of our model (S1 Text, Section 6.2; S3 Fig).

**Quantifying the deviation from statistical independence.** For $R_{0,i} > 1$, the equilibrium prevalence of coinfection in our deterministic model is given by

$$\bar{J}_{1,2} = \left( \frac{\beta_1 + \beta_2}{\beta_1 + \beta_2 - \mu} \right) \bar{I}_1 \bar{I}_2. \tag{4}$$

(See also Methods section 'Equilibria of the two-pathogen model'). We introduce $\Lambda$, the relative deviation of the prevalence of coinfection from that required by statistical independence ($\bar{P} = \bar{I}_1 \bar{I}_2$), which here is given by

$$\Lambda = \frac{\bar{J}_{1,2} - \bar{P}}{\bar{P}} = \frac{\mu}{\beta_1 + \beta_2 - \mu} = \frac{1}{R_{0,1} + R_{0,2} - 1} \geq 0. \tag{5}$$

The deviation is zero if, and only if, the host natural death rate is $\mu = 0$. The observed outcome would therefore conform with statistical independence only for noninteracting pathogens when there is no host natural death (at the time scale of an infection). This reiterates the role of host natural death in causing deviation from a statistical association pattern. The relative deviation from statistical independence, $\Lambda$, becomes smaller as either or both values of $R_{0,i}$ become larger. Deviations are therefore more apparent for smaller values of $R_{0,i}$. This is unsurprising, since if either pathogen has a very large value of $R_0$, almost all hosts infected with the other pathogen would be expected to become coinfected, and so both our model and the assumption of statistical independence would lead to very similar predictions.

This result (Eq 5) was first published by Kucharski and Gog [32] in a different context (model reduction in multistrain influenza models). Moreover, using a continuous age-structured model, these authors showed that one may recover statistical independence within infinitesimal age classes. The result in Eq 5 is related to ageing, as individuals acquire more infections as they age. As age increases, so does the probability of being infected with pathogens 1 and/or

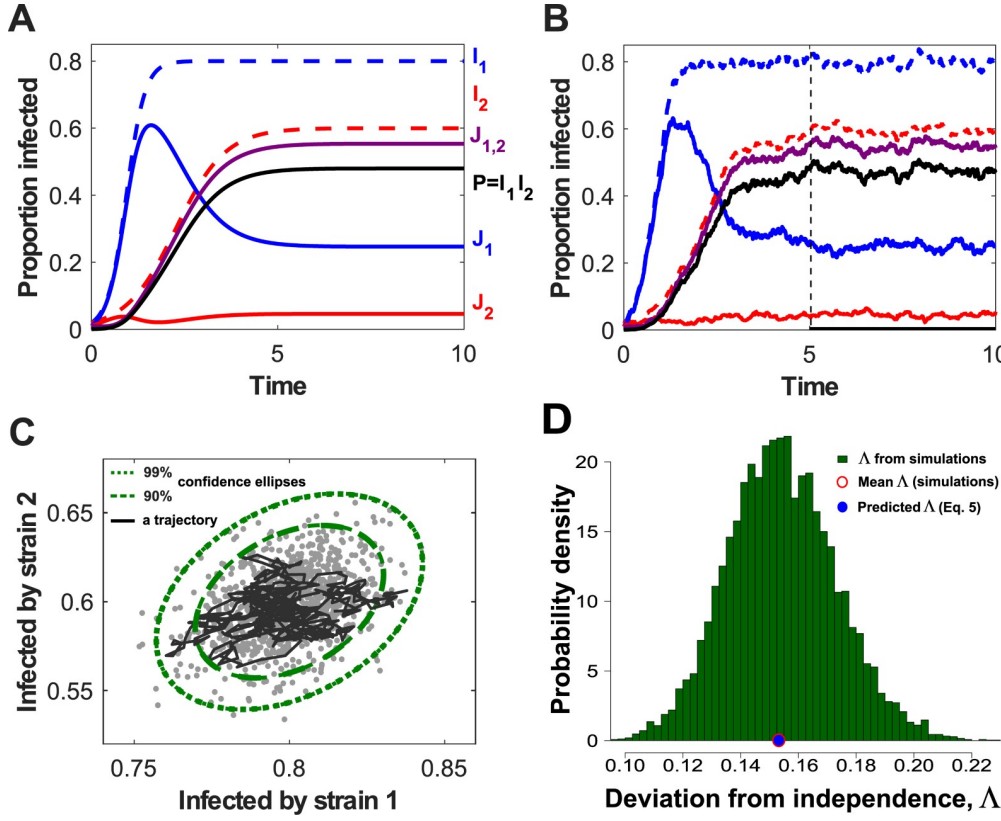

**Fig 2. Simulations of the two-pathogen model show that net densities of the two pathogens are positively correlated.** $J_1$ and $J_2$ are hosts singly infected by pathogens 1 and 2, respectively; $J_{1,2}$ is coinfected hosts; $I_1 = J_1 + J_{1,2}$ and $I_2 = J_2 + J_{1,2}$ are net densities of hosts infected by pathogens 1 and 2, respectively. (A) Dynamics of the deterministic model (Eqs 1–3), with $\beta_1 = 5$, $\beta_2 = 2.5$, and $\mu = 1$ (parameters have units of inverse time). (B) Dynamics of a stochastic version of the model, in a population of size $N = 1,000$ (see also Methods section 'Stochastic models'). (C) A single trajectory from the stochastic simulation (black line) in panel B (restricted to the time interval starting from the dashed line at $t = 5$) in the phase plane $(I_1, I_2)$, and the 90% and 99% confidence ellipses (dashed and dotted curves, respectively) generated from an analytical approximation to the stochastic model. The grey dots represent pairs of individual values of $(I_1, I_2)$ sampled at time $T = 10$ from $10^3$ independent replicates of the stochastic model. (D) The probability density of $\Lambda$, the relative deviation of the prevalence of coinfection (Eq 5), as estimated at $T = 10$ from $10^4$ independent replicates of our stochastic model when $N = 1,000$. The relative deviation was much greater than zero in all individual simulations. The mean value of $\Lambda$ as estimated from these simulations, $\bar{\Lambda} \approx 0.1542$, was extremely close to the prediction from the deterministic model, $\Lambda = 1/(5 + 2.5 - 1) \approx 0.1538$ (Eq 5; these values are marked by red open and blue filled dots on the x-axis).

2. Therefore, the prevalences of pathogens 1 and 2 are positively correlated [33]. A greater deviation from independence as the mortality rate $\mu$ increases is likely due to the fact that prevalence is increasing and concave with respect to age and saturates in older age classes [31].

## Testing for interactions between pathogens

Eq 3 can be straightforwardly extended to track $n$ pathogens that do not interact in any way (including pairwise and three-way interactions). Equilibria of this model are prevalences of different classes of infected or coinfected hosts carrying different combinations of noninteracting pathogens. These can be used to derive a test for interaction between pathogens that properly accounts for the lack of statistical independence revealed by our analysis of the simple two-pathogen model.

**Modelling coinfection by $n$ noninteracting pathogens.** We denote the proportion of hosts simultaneously coinfected by the (nonempty) set of pathogens $\Gamma$ to be $J_\Gamma$ and use $\Omega_i = \Gamma \setminus \{i\}$ (for $i \in \Gamma$) to represent combinations with one fewer pathogen.

The dynamics of the $2^n - 1$ distinct values of $J_\Gamma$ follow

$$\dot{J}_\Gamma = \sum_{i \in \Gamma} F_i J_{\Omega_i} - \left( \sum_{i \notin \Gamma} F_i + \mu \right) J_\Gamma, \tag{6}$$

in which the net force of infection of pathogen $i$ is

$$F_i = \beta_i I_i = \beta_i \sum_{\Gamma \in \nabla_i} J_\Gamma, \tag{7}$$

and $\nabla_i$ is the set of all subsets of $\{1,\ldots,n\}$ containing $i$ as an element. Eq 6 can be interpreted by noting the following:

- the first term tracks inflow due to hosts carrying one fewer pathogen becoming infected;

- the second term tracks the outflows due to hosts becoming infected by an additional pathogen, or death.

If $R_{0,i} = \beta_i/\mu > 1$ for all $i = 1,\ldots,n$, the equilibrium prevalence of hosts predicted to be infected by any given combination of pathogens, $\bar{J}_\Gamma$, can be obtained by (recursively) solving a system of $2^n$ linear equations (Eq 16 in Methods section 'Equilibria of the $n$-pathogen model').

These equilibrium prevalences are the prediction of our 'Noninteracting Distinct Pathogens' (NiDP) model, which in dimensionless form has $n$ parameters (the $R_{0,i}$'s, $i = 1,\ldots,n$; Methods section 'Fitting the models').

If we simplify the model by assuming that all pathogens are epidemiologically interchangeable and so all pathogen infection rates are equal (i.e., $\beta_i = \beta$ for all $i$), then if $R_0 = \beta/\mu > 1$, the proportion of hosts infected by $k$ distinct pathogens can be obtained by (recursively) solving $n +1$ linear equations (Eq 22 in Methods section 'Deriving the NiSP model from the NiDP model'). This constitutes the prediction of our 'Noninteracting Similar Pathogens' (NiSP) model, a simplified form of the NiDP model requiring only a single parameter ($R_0$).

**Using the models to test for interactions.** If either the NiSP or NiDP model adequately explains coinfection data, those data are consistent with the underpinning assumption that pathogens do not interact. Which model is fitted depends on the form of the available data, specifically whether only the number of pathogens or instead which particular combination of pathogens infecting each host is known.

Studies often quantify only the number of distinct pathogens carried by individual hosts, without necessarily specifying the combinations involved [22, 45–50]. There are insufficient degrees of freedom in such data to fit the NiDP model, and so we fall back upon the NiSP model. In using the NiSP model, we additionally assume all pathogens within a given study are epidemiologically interchangeable.

We identified four suitable studies reporting data concerning strains/clones of a single pathogen and tested whether these data are consistent with no interaction. For all four studies (Fig 3), the best-fitting NiSP model is a better fit to the data than the corresponding binomial model assuming statistical independence (Eq 28 in Methods section 'Models corresponding to assuming statistical independence'). Application of our model to three additional examples for data sets considering distinct pathogens, which deviate more markedly from the epidemiological equivalence assumption, is described in S2 Text, Section 1 (see also S4 Fig, S1 Table, S2 Table).

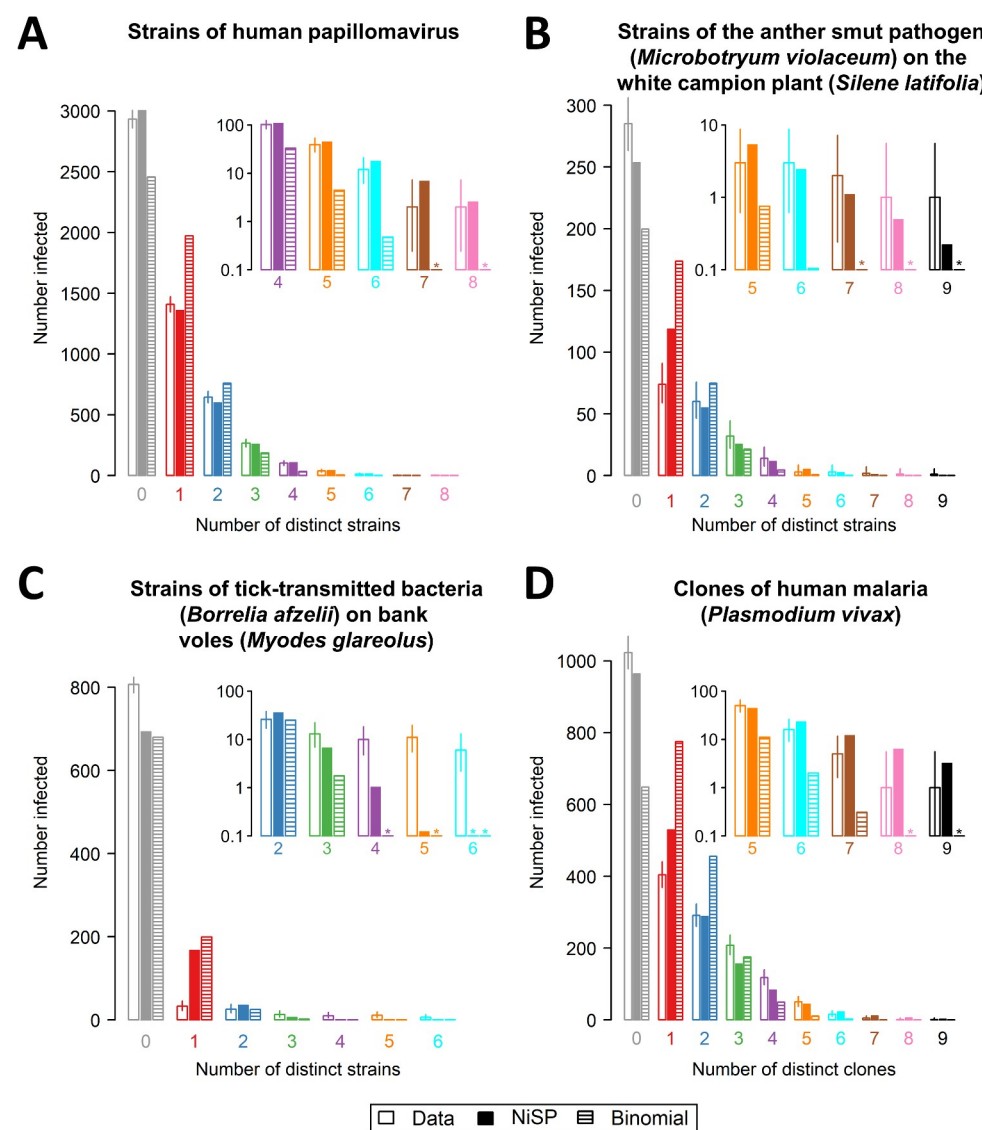

**Fig 3. Comparing predictions of the NiSP model with binomial models assuming statistical independence.** In using the NiSP model, pathogens are assumed to be epidemiologically interchangeable: we have therefore restricted attention to data sets concerning strains/clones of a single pathogen species. (A) Strains of human papillomavirus [22]; (B) strains of the anther smut pathogen (*Microbotryum violaceum*) on the white campion (*Silene latifolia*) [45]; (C) strains of tick-transmitted bacteria (*Borrelia afzelii*) on bank voles (*Myodes glareolus*) [46]; and (D) clones of malaria (*Plasmodium vivax*) [47]. Insets to each panel show a 'zoomed-in' section of the graph corresponding to high multiplicities of clone/strain coinfection, using a logarithmic scale on the y-axis for clarity. Asterisks indicate predicted counts smaller than 0.1. In all four cases, the NiSP model is a better fit to the data than the binomial model (Δ AIC = 572.8,158.6,293.8 and 596.3, respectively). For the data shown in (A), there is no evidence that the NiSP model does not fit the data (lack of goodness of fit *p* = 0.08), and so our test indicates the human papillomavirus strains do not interact. For the data shown in (B–D), there is evidence of lack of goodness of fit (all have lack of goodness of fit *p*<0.01). Our test therefore indicates these strains/clones interact (or are epidemiologically different). The underlying data for this figure can be found in S3 Data, S4 Data, S5 Data, and S6 Data. AIC, Akaike information criterion; NiSP, Noninteracting Similar Pathogens.

In one case—coinfection by different strains of human papillomavirus (HPV) [22] (Fig 3A)— we find no evidence that the reported data cannot be explained by the NiSP model. These data therefore support the hypothesis of no interaction—and indeed no epidemiological differences— between the pathogen strains in question.

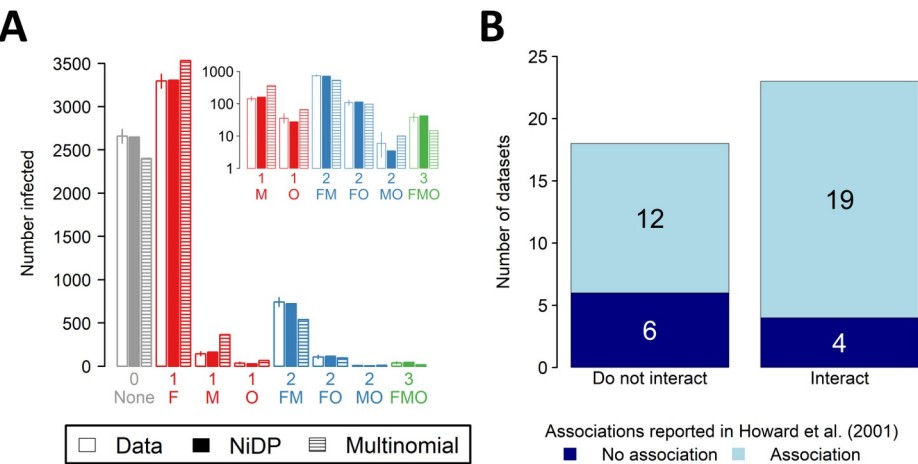

**Fig 4. Using the NiDP model to reanalyse malaria data sets considered by Howard and colleagues [17].** In using the NiDP model, there is no need to assume malaria-causing *Plasmodium* spp. are epidemiologically interchangeable. (A) Comparing the predictions of the NiDP model with a multinomial model of infection (i.e., statistical independence) for the data set on *P. falciparum* ('F'), *P. malariae* ('M'), and *P. ovale* ('O') coinfection in Nigeria reported by Molineaux and colleagues [51]. The NiDP model is a better fit to the data than the multinomial model (Δ AIC = 326.2); additionally, there is no evidence of lack of goodness of fit (*p* = 0.40). This data set is therefore consistent with no interaction between the three *Plasmodium* species. (B) Comparing the results of fitting the NiDP model and the methodology of Howard and colleagues [17] based on log-linear regression and so statistical independence. For 16 (i.e., 12 + 4) out of the 41 data sets we considered, the conclusions of the two methods differ. The underlying data for this figure can be found in S7 Data and S8 Data. AIC, Akaike information criterion; NiDP, Noninteracting Distinct Pathogens.

In the three other cases we considered—strains of anther smut (*M. violaceum*) on the white campion (*S. latifolia*) [45] (Fig 3B), strains of the tick-transmitted bacterium *B. afzelii* on bank voles (*M. glareolus*) [46] (Fig 3C), and clones of a single malaria parasite (*P. vivax*) infecting children [47] (Fig 3D)—despite outperforming the model corresponding to statistical independence, the best-fitting NiSP model does not adequately explain the data. We therefore reject the hypotheses of no interaction in all three cases, noting that our use of the NiSP model means it might be epidemiological differences between pathogen strains/clones—or perhaps simply lack of fit of the underpinning S-I-S model—that have in fact been revealed.

Other studies report the proportion of hosts infected by particular combinations (rather than counts) of pathogens, although many of those concentrate on helminth macroparasites for which our underlying S-I-S model is well known to be inappropriate [54].

However, a methodological article by Howard and colleagues [17] introduces the use of log-linear modelling to test for statistical associations. Conveniently, that article reports the results of that methodology as applied to a large number of studies focusing on *Plasmodium* spp. causing malaria.

By interrogating the original data sources (Methods section 'Combinations of pathogens [NiDP model]'), we found a total of 41 studies of malaria reporting the disease status of at least *N* = 100 individuals, and in which three of *P. falciparum*, *P. malariae*, *P. ovale*, and *P. vivax* were considered. Data therefore consist of counts of the number of individuals infected with different combinations of three of these four pathogens, a total of eight classes. There were sufficient degrees of freedom to fit the NiDP model, which here has three parameters, each corresponding to the infection rate of a single *Plasmodium* spp. Fig 4A shows the example of fitting the NiDP model to data from a study of malaria in Nigeria [51].

Fitting the NiDP model allows us to test for interactions between *Plasmodium* spp., without assuming they are epidemiologically interchangeable. In 18 of the 41 cases we considered, our

methods suggest the data are consistent with no interaction (Fig 4B). We note that in 12 of these 18 cases, the methodology based on statistical independence of Howard and colleagues [17] instead suggests the *Plasmodium* spp. interact.

## Discussion

We have shown that pathogens that do not interact and so have uncoupled prevalence dynamics (Eq 1) are not statistically independent. For two pathogens, the prevalence of coinfection is always greater than the product of the prevalences (Eq 5), unless host natural death does not occur. This result was first published in an age-structured, multistrain influenza model [32]. Pathogens share a single host in coinfections, and so when a coinfected host dies, net prevalences of both pathogens decrease simultaneously. The prevalences of individual pathogens, regarded as random variables, therefore covary positively. A related interpretation is due to Kucharski and Gog [32]: the prevalences of the pathogens are positively correlated through a single independent variable, namely the age of the hosts. As a side result, we note our analysis indicates that a high-profile, oft-cited model of May and Nowak [55] is based on a faulty assumption of probabilistic independence (S1 Text, Section 3). More importantly, our analysis also shows that statistically independent pathogens may well be interacting (S1 Text, Section 5), which confirms that statistical independence is far from equivalent to the absence of biological interaction between pathogens.

More specifically, our results highlight that positive correlations between densities of infected hosts are a reasonable expectation, even if the pathogens in question do not interact. It might even be that a positive correlation is found when there is in fact a negative interaction, providing the confounding effect of age is sufficiently strong. In this context, results concerning the reliability of detecting associations between nematodes and intestinal coccidia (*Eimeria* spp.) in natural small-mammal populations presented by Fenton and colleagues [14] are notable. These authors found that correlation-based cross-sectional analyses often revealed positive associations between pairs of parasites known to interact negatively with each other (Fig 2 in [14]). Although our S-I-S model—strictly speaking—is not applicable to macroparasites, including nematodes, it might be that our results can provide a partial theoretical explanation of these findings (see also [56], which reports a relative overabundance of positive associations between resident parasites of 22 small-mammal species). Testing whether and how our methods generalise to macroparasites would be an interesting development of the work presented here, and it is possible that such a modelling exercise would provide a theoretical context to understand these types of correlations in macroparasite data.

We extended our model to an arbitrary number of pathogens to develop a novel test for interaction that properly accounts for statistical nonindependence. Many data sets summarise coinfections in terms of multiplicity of infection, regardless of which pathogens are involved. Since there would then be as many epidemiological parameters as pathogens in our default NiDP model, and so as many parameters as data points, the full model would be overparameterised. We therefore introduced the additional assumption that all pathogens are epidemiologically interchangeable. This formed the basis of the parsimonious NiSP model, which is most appropriate for testing for interactions between strains or clones of a single pathogen species.

Despite the strong and perhaps even unrealistic assumption that strains/clones are interchangeable, the NiSP model outperformed the binomial model assuming statistical independence for all four data sets we considered. In particular, the NiSP model successfully captured the fat tails characteristic of observed multiplicity of infection distributions. All four data sets

therefore support the idea that coinfection is far more frequent than statistical independence would imply.

For the data set concerning coinfection by different strains of HPV [22], the NiSP model also passed a goodness of fit test, allowing us to conclude strains of this pathogen do not interact. Goodness of fit for such a simple model is a particularly conservative test, especially for the NiSP model, in which we assume pathogens clones/strains are epidemiologically interchangeable. However, our analysis relied on a model reflecting a natural history most suitable for chronic infections, with hosts infected until they die. In contrast, HPV infections may be acute, meaning there is clearance within a few years [42]. However, extending our model to handle pathogen-specific rates of clearance (S1 Text, Section 4) yielded qualitatively similar results (S2 Text, Section 3; S5 Fig), suggesting the difference in natural history between HPV and our model is not driving the results obtained here.

More generally, we wish to very explicitly highlight that here, we illustrated our methods via case studies for which suitable data are readily available, and our purpose was not to come to definitive conclusions concerning any particular system. That would require dedicated studies. However, by fitting even a highly simplified version of our model to data, we have demonstrated how results of simple epidemiological models challenge previous methods based on statistical independence.

To explore further the implications of our findings, we analysed available data sets, tracking distinct combinations of pathogens involved in each occurrence of coinfection. For methodological-comparison purposes, we restricted ourselves to data referenced by Howard and colleagues [17] concerning interactions between *Plasmodium* spp. causing malaria. Relaxing the assumption of epidemiological interchangeability (i.e., using the NiDP model), we found that 43.9% (i.e., 18/41) of data sets considered by Howard and colleagues [17] are consistent with no interaction.

One may wonder whether focusing on age classes may be sufficient to correct for the positive correlation between noninteracting pathogens [31]. Of the 41 data sets identified by Howard and colleagues [17] that we analysed, 14 focused only on data collected from children, and therefore, associations are less likely to emerge solely by the confounding effect of age [35]. Of these 14 studies, we came to the same conclusion as Howard and colleagues [17] in only six cases. We identified two cases in which our methods suggest there is an interaction in which Howard and colleagues [17] concluded no interaction (studies 71 and 77), as well as six cases in which we conclude no interaction, whereas Howard and colleagues [17] conclude there is an interaction (studies 76, 68, 69, 70, 79, and 80). Thus, focusing on discrete and arbitrary age classes may not be sufficient to correct for the positive correlation between noninteracting pathogens.

Again, we do not intend to conclusively demonstrate interactions—or lack of interactions—for malaria. Instead, what is important is that our results very often diverge from those originally reported by Howard and colleagues [17] using a method based on statistical associations, namely log-linear regression. Log-linear regression suffers from well-acknowledged difficulties in cases in which there are zero counts (i.e., certain combinations of pathogens are not observed) [57]. Such cases often arise in epidemiology. Methods based on epidemiological models therefore offer a 2-fold advantage: biological interactions are not confounded with statistical associations, and parameter estimation is well posed, irrespective of zero counts.

Moreover, simple epidemiological models (with no explicit age structure) intrinsically correct the bias due to the positive correlation between age and prevalence, which makes it unnecessary to control for age. Therefore (and this may be our main conclusion), although age is an evident confounding factor, epidemiological models make it unnecessary to keep track of the age of infected hosts. This is made possible by replacing the paradigm of 'statistical

independence and random distributions' with 'model-based distributions in absence of biological interactions'.

We focused here on the simple S-I-S model, since it is sufficiently generic to be applicable to a number of systems. However, an important assumption of our model—as discussed earlier for the case of HPV—is that natural mortality occurs at a time scale comparable to that of an infection. Our model is therefore tailored for chronic (i.e., long-lasting) infections, although we note this represents a large fraction of coinfections in humans, animals, and (particularly) plants. Also, our study is restricted to nonlethal infections, as otherwise there may be ecological interactions between pathogens [30]. In future work focusing on interactions between particular pathogens, models including additional system-specific detail would, of course, be appropriate.

Other work could also perhaps explicitly focus on more complex epidemiological dynamics that are relevant across an entire range of systems. As previously discussed, one possibility is of extending the work to include an underlying model that better represents macroparasite dynamics [58]. Another example is simultaneous transmission of multiple pathogens/strains/clones, which can be particularly relevant in the case of chronic viruses [59]. There are other aspects of host–pathogen interactions that can be important for the dynamics of chronic infections but that are omitted in the simple S-I-S type model. An obvious example is spatially explicit transmission rates, often represented in models by way of contact networks [60] or individual-based models at small [61] or large spatial scales [62] or the meta-population paradigm [63]. Another factor currently omitted is individual-level heterogeneity in transmission and/or susceptibility, for example, due to immunity [64] or genetic variation within host populations [65]. In principle, our methods could be extended by including these or any other heterogeneities in the underlying model and testing the extent to which such a model can explain observed prevalence data when pathogens are assumed not to interact. However, the difficulty would then be in model parameterisation, since it is unlikely that such complex models could straightforwardly be fitted using the type of cross-sectional data used here; but see Alizon and colleagues [42] for parameter inference using Approximate Bayesian Computation in a model including host heterogeneity in behaviour. Consequently, we defer further analysis of these and other more complex underpinning epidemiological models to future research.

We also focused here on tests based on the equilibrium behaviour of our models. In practice, coinfecting pathogen populations might not yet have equilibrated within the host population of interest, and so it is possible that transient dynamics might play an important role. However, for tests to be used with cross-sectional data, the assumption of equilibrium is a logical point of departure. If longitudinal data were to be available, the methodology presented here could be extended in the obvious way, i.e., by explicitly fitting a compartmental model in which pathogens do not interact to prevalence data collected at multiple time points (see, e.g., [40]). An initial investigation of the first time at which the prevalence of coinfection exceeds the product of prevalences in our simplest two-pathogen model suggests that it is often the case that the naïve prediction based on statistical independence becomes invalid relatively quickly, within the average lifetime of a single host. This remains the case for a wide variety of parameter sets and initial conditions (S1 Text, Section 1.2; S1 Fig). At least for our default parameter set (i.e., the parameters used in Fig 2), irrespective of the initial condition, simulations of our stochastic model also indicate that any other transient behaviour is also 'washed out' of the system relatively quickly. The full relative deviation from statistical independence is therefore quickly attained irrespective of the particular initial conditions (see also S1 Text, Section 6.2; S3 Fig). However, we defer more in-depth

analysis of transient behaviour, including attempting to characterise when any signal might be detectable from real data, to future work.

Lastly, we speculate our results may have implications beyond epidemiology. After all, pathogens are species that form meta-populations occupying discrete patches (hosts) [66]. Meta-community ecology has long been concerned with whether interactions between species can be detected from co-occurrence data [43, 67, 68], and most existing methods are based on detecting statistical associations [69, 70] (but see also Hastings [71]). We therefore simply highlight that our dynamical modelling approach may also provide a new perspective in this area.

## Methods

### Mathematical analyses

**Equilibria of the two-pathogen model.** The two-pathogen model is given by Eqs 1–3. Since the population size is constant, $J_\varnothing = 1 - J_1 - J_2 - J_{1,2}$, and so it follows that

$$\dot{J}_\varnothing = \mu(J_1 + J_2 + J_{1,2}) - (F_1 + F_2)J_\varnothing = \mu(1 - J_\varnothing) - (F_1 + F_2)J_\varnothing. \tag{8}$$

It is well known [52] that if $R_{0,i} = \beta_i/\mu > 1$ and $I_i(0) > 0$, the prevalence of pathogen $i$ will tend to an equilibrium $\bar{I}_i = 1 - 1/R_{0,i}$.

Since $F_i = \beta_i I_i$ and $J_i = I_i - J_{1,2}$, the rate of change of coinfected hosts in Eq 3 can be recast as

$$\dot{J}_{1,2} = \beta_2 I_2(I_1 - J_{1,2}) + \beta_1 I_1(I_2 - J_{1,2}) - \mu J_{1,2}, \tag{9}$$

which leads immediately to the results given in Eqs 4 and 5.

**Equilibria of the $n$-pathogen model.** The $n$-pathogen model is given by Eqs 1, 6 and 7. Since the host population size is constant, $J_\varnothing = 1 - \sum_{\Gamma \in \nabla} J_\Gamma$, where $\nabla$ is the set of all $2^n - 1$ sets with infected or coinfected hosts. It is also true that

$$\dot{J}_\varnothing = \mu(1 - J_\varnothing) - \left(\sum_{i=1}^{n} F_i\right) J_\varnothing. \tag{10}$$

At equilibrium, Eq 6 becomes

$$0 = \sum_{i \in \Gamma} \bar{F}_i \bar{J}_{\Omega_i} - \left(\sum_{i \notin \Gamma} \bar{F}_i + \mu\right) \bar{J}_\Gamma, \tag{11}$$

in which $\bar{J}_{\Omega_i}$ and $\bar{J}_\Gamma$ are equilibrium prevalences, and $\bar{F}_i$ is the force of infection of pathogen $i$ at equilibrium, i.e.,

$$\bar{F}_i = \beta_i \bar{I}_i = \beta_i\left(1 - \frac{\mu}{\beta_i}\right) = \beta_i - \mu. \tag{12}$$

Since these forces of infection are constant and do not depend on the equilibrium prevalences, the set of $2^n - 1$ equations partially characterising the equilibrium is linear, with

$$0 = \sum_{i \in \Gamma} (\beta_i - \mu) \bar{J}_{\Omega_i} - \left(\sum_{i \notin \Gamma} (\beta_i - \mu) + \mu\right) \bar{J}_\Gamma. \tag{13}$$

Similarly, Eq 10 is linear

$$0 = \mu(1 - \bar{J}_\varnothing) - \left( \sum_{i=1}^{n} (\beta_i - \mu) \right) \bar{J}_\varnothing. \tag{14}$$

The equilibrium prevalences can be written very conveniently in a recursive form (i.e., using the first equation to fix $\bar{J}_\varnothing$, using $\bar{J}_\varnothing$ to independently calculate all values of $\bar{J}_\Gamma$ for $|\Gamma| = 1$, then using the set of values of $\bar{J}_\Gamma$ when $|\Gamma| = 1$ to independently calculate all values of $\bar{J}_\Gamma$ for $|\Gamma| = 2$, and so on). A recurrence relation to find all equilibrium prevalences can therefore be initiated with the following expression for the density of uninfected hosts:

$$\bar{J}_\varnothing = \frac{\mu}{\mu + \sum_{i=1}^{n} (\beta_i - \mu)} = \frac{1}{1 + \sum_{i=1}^{n} (R_{0,i} - 1)}. \tag{15}$$

Then, one may recursively use the following equation, equivalent to Eq 13:

$$\bar{J}_\Gamma = \frac{\sum_{i \in \Gamma} (\beta_i - \mu) \bar{J}_{\Omega_i}}{\mu + \sum_{i \notin \Gamma} (\beta_i - \mu)} = \frac{\sum_{i \in \Gamma} (R_{0,i} - 1) \bar{J}_{\Omega_i}}{1 + \sum_{i \notin \Gamma} (R_{0,i} - 1)}. \tag{16}$$

Since the densities in Eq 16 are entirely in terms of the equilibrium densities of hosts carrying one fewer pathogen ($\bar{J}_{\Omega_i}$), this allows us to recursively find the densities of all pathogens given pathogen-by-pathogen values of $R_{0,i}$.

**Deriving the NiSP model from the NiDP model.** If all pathogens are interchangeable and so have identical values of $R_{0,i} = R_0 \ \forall i$, then for any pair of combinations of infecting pathogens, $\Gamma_1$ and $\Gamma_2$, it must be the case that $\bar{J}_{\Gamma_1} = \bar{J}_{\Gamma_2}$ whenever $|\Gamma_1| = |\Gamma_2|$. This means the equilibrium prevalences of hosts infected by the same number of distinct pathogens must all be equal, irrespective of the particular combination of pathogens that is carried. In this case, solving the system is much simpler. First, Eq 11 can be rewritten as

$$0 = |\Gamma| \bar{F} \bar{J}_{\Omega_i} - ((n - |\Gamma|) \bar{F} + \mu) \bar{J}_\Gamma, \tag{17}$$

in which $\bar{F} = \beta - \mu$. The net prevalence of hosts infected by $k$ distinct pathogens is

$$\bar{M}_k = \sum_{\Gamma \in \nabla(k)} \bar{J}_\Gamma, \tag{18}$$

in which $\nabla(k)$ is the set of combinations of $\{1, \ldots, n\}$ with $k$ elements. Since the form of Eq 17 depends only on $|\Gamma|$, all individual prevalences involved in $\bar{M}_k$ are identical, and so

$$\bar{M}_k = C_k^n \bar{J}_{\Gamma,k}, \tag{19}$$

in which $C_k^n$ is a combinatorial coefficient, and $\bar{J}_{\Gamma,k}$ is any of the individual prevalences for which $|\Gamma| = k$. The ratio between successive values of $\bar{M}_k$ is given by

$$\frac{\bar{M}_k}{\bar{M}_{k-1}} = \frac{C_k^n}{C_{k-1}^n} \frac{\bar{J}_{\Gamma,k}}{\bar{J}_{\Gamma,k-1}} = \frac{n - k + 1}{k} \frac{\bar{J}_{\Gamma,k}}{\bar{J}_{\Gamma,k-1}}. \tag{20}$$

From Eq 15, it follows that

$$\bar{M}_0 = \frac{\mu}{\mu + n\bar{F}} = \frac{1}{1 + n(R_0 - 1)}, \tag{21}$$

in which $R_0 = \beta/\mu$. For $1 \leq k \leq n$, Eqs 17 and 20 together imply

$$\bar{M}_k = \frac{(n - k + 1)\bar{F}}{(n - k)\bar{F} + \mu} \bar{M}_{k-1} = \frac{(n - k + 1)(R_0 - 1)}{(n - k)(R_0 - 1) + 1} \bar{M}_{k-1}, \tag{22}$$

a form that admits a simple recursive solution.

**Stochastic models.** Fig 2B and 2C were generated by simulating the stochastic differential equation (SDE) corresponding to Eq 3, simulating a continuous-time Markov chain model using Gillespie's algorithm gave consistent results. Confidence ellipses were obtained from an approximate expression for the covariance matrix at equilibrium (see below).

The continuous-time Markov chain model corresponding to the unscaled version of Eqs 3–8 tracks a vector of integer-valued random variables $X(t) = (J_\varnothing(t), J_1(t), J_2(t), J_{1,2}(t))$. Defining $\Delta X = X(t+\Delta t) - X(t) = (\Delta J_\varnothing, \Delta J_1, \Delta J_2, \Delta J_{1,2})$, changes of $\pm 1$ to each element of $X(t)$ occur in small periods of time $\Delta t$ at the rates given in Table 1. Stochastic trajectories from this model can conveniently be simulated via the Gillespie algorithm [72]. Note that the numeric values of the infection rates and the host birth rate must be altered to account for the scaling by population size.

The model can also be written as a system of SDEs, an approximation to the continuous-time Markov chain that is valid for sufficiently large $N$ [73] and that is particularly well suited for simulation of the stochastic model when the population size is large. This form of the model again tracks the seven events in Table 1, although in the SDE formulation the random variables in $X(t)$ are continuous-valued. A heuristic derivation is based on a normal approximation described below. Alternately, the forward Kolmogorov differential equations in the continuous-time Markov chain model are closely related to the Fokker-Planck equation for the probability density function of the SDE model [74].

**Table 1. Transitions in the two-pathogen stochastic models.** The prevalence of uninfected host is $J_\varnothing$, the prevalence of each class of singly infected hosts is $J_i$ (for $i \in [1,2]$), and the prevalence of coinfected host is $J_{1,2}$. The net force of infection of pathogen $i$ is $F_i = \beta_i I_i/N = \beta_i(J_i + J_{1,2})/N$ (note the scaling by the population size $N$ relative to the forces of infection as used in the deterministic version of the model). To ensure a constant host population size, we have made the simplifying assumption that removal and replacement occur simultaneously; this has no effect on our qualitative results.

| Event Number | Event | Rate | Change(s) to state variable(s) ($\Delta X$) |
|---|---|---|---|
| 1 | Infection of uninfected host by pathogen 1 | $F_1 J_\varnothing \Delta t + o(\Delta t)$ | $J_\varnothing \rightarrow J_\varnothing - 1$ $J_1 \rightarrow J_1 + 1$ |
| 2 | Infection of uninfected host by pathogen 2 | $F_2 J_\varnothing \Delta t + o(\Delta t)$ | $J_\varnothing \rightarrow J_\varnothing - 1$ $J_2 \rightarrow J_2 + 1$ |
| 3 | Infection by pathogen 1 of host singly infected by pathogen 2 | $F_1 J_2 \Delta t + o(\Delta t)$ | $J_2 \rightarrow J_2 - 1$ $J_{1,2} \rightarrow J_{1,2} + 1$ |
| 4 | Infection by pathogen 2 of host singly infected by pathogen 1 | $F_2 J_1 \Delta t + o(\Delta t)$ | $J_1 \rightarrow J_1 - 1$ $J_{1,2} \rightarrow J_{1,2} + 1$ |
| 5 | Death of host singly infected by pathogen 1 and replacement with an uninfected host | $\mu J_1 \Delta t + o(\Delta t)$ | $J_1 \rightarrow J_1 - 1$ $J_\varnothing \rightarrow J_\varnothing + 1$ |
| 6 | Death of host singly infected by pathogen 2 and replacement with an uninfected host | $\mu J_2 \Delta t + o(\Delta t)$ | $J_2 \rightarrow J_2 - 1$ $J_\varnothing \rightarrow J_\varnothing + 1$ |
| 7 | Death of coinfected host and replacement with an uninfected host | $\mu J_{1,2} \Delta t + o(\Delta t)$ | $J_{1,2} \rightarrow J_{1,2} - 1$ $J_\varnothing \rightarrow J_\varnothing + 1$ |

The expected change $\mathbb{E}(\Delta X)$ and covariance of the changes $\mathbb{V}(\Delta X)$ can be computed from Table 1 to order $\Delta t$ via

$$\mathbb{E}(\Delta X) \approx \tilde{f}\Delta t \text{ and } \mathbb{V}(\Delta X) \approx \mathbb{E}(\Delta X[\Delta X]^T) = \Sigma\Delta t, \quad (23)$$

where $dJ = \tilde{f}dt$ is the unscaled version of the deterministic model as specified in Eqs 3–8 with $N = J_\varnothing + J_1 + J_2 + J_{1,2}$ (a constant) and $F_i = \beta_i(J_i + J_{1,2})/N$. In addition, the matrix $\Sigma$ is given by

$$\begin{bmatrix} \mu(N - J_\varnothing) + (F_1 + F_2)J_\varnothing & -F_1 J_\varnothing - \mu J_1 & -F_2 J_\varnothing - \mu J_2 & -\mu J_{1,2} \\ -F_1 J_\varnothing - \mu J_1 & F_1 J_\varnothing + (F_2 + \mu)J_1 & 0 & -F_2 J_1 \\ -F_2 J_\varnothing - \mu J_2 & 0 & F_2 J_\varnothing + (F_1 + \mu)J_2 & -F_1 J_2 \\ -\mu J_{1,2} & -F_2 J_1 & -F_1 J_2 & F_2 J_1 + F_1 J_2 + \mu J_{1,2} \end{bmatrix} \quad (24)$$

The changes in a small time interval $\Delta t$ are approximated by a normal distribution via the central limit theorem: $\Delta X(t) - \mathbb{E}(\Delta X(t)) \approx \text{Normal}(0, \Sigma\Delta t)$, where $0$ = zero vector. The covariance matrix $\Sigma$ can be written as $\Sigma = GG^T$. Letting $\Delta t \to 0$, the SDE model can therefore be expressed as

$$dX = \tilde{f}dt + GdW. \quad (25)$$

The matrix $G$ is not unique, but a simple form with dimension 4×7 accounts for each event in Table 1 [74]. Each entry in matrix $G$ involves a square root, and $W$ is a vector of seven independent standard Wiener processes, where $dW_i \approx \Delta W_i(t) = W_i(t+\Delta t) - W_i(t) \sim \text{Normal}(0, \Delta t)$. An explicit form for the SDE model in Eq 25 is

$$\begin{aligned} dJ_\varnothing &= \tilde{f}_0 dt - \sqrt{F_1 J_\varnothing}dW_1 - \sqrt{F_2 J_\varnothing}dW_2 + \sqrt{\mu J_1}dW_5 + \sqrt{\mu J_2}dW_6 + \sqrt{\mu J_{1,2}}dW_7, \\ dJ_1 &= \tilde{f}_1 dt + \sqrt{F_1 J_\varnothing}dW_1 - \sqrt{F_2 J_1}dW_4 - \sqrt{\mu J_1}dW_5, \\ dJ_2 &= \tilde{f}_2 dt + \sqrt{F_2 J_\varnothing}dW_2 - \sqrt{F_1 J_2}dW_3 - \sqrt{\mu J_2}dW_6, \\ dJ_{1,2} &= \tilde{f}_{1,2} dt + \sqrt{F_1 J_2}dW_3 + \sqrt{F_2 J_1}dW_4 - \sqrt{\mu J_{1,2}}dW_7. \end{aligned} \quad (26)$$

In S1 Text, Section 2, we show that the covariance between the prevalences of pathogen 1 and pathogen 2 as they fluctuate in the vicinity of their equilibrium values is approximately

$$\text{cov}\left(\frac{I_1}{N}, \frac{I_2}{N}\right) = \frac{\mu \bar{J}_{1,2}}{N^2[(\beta_1 - \mu) + (\beta_2 - \mu)]} = \frac{(\beta_1 + \beta_2)(\beta_1 - \mu)(\beta_2 - \mu)\mu}{N\beta_1\beta_2(\beta_1 + \beta_2 - \mu)(\beta_1 - \mu + \beta_2 - \mu)} \geq 0, \quad (27)$$

with equality if, and only if, $\mu = 0$ (assuming $\beta_i > \mu$, $i = 1,2$). Only in the specific case $\mu = 0$ is the deviation from statistical independence equal to zero (Eq 5).

## Statistical methods

**Models corresponding to assuming statistical independence.** If data are observations of numbers of individuals infected with $k$ distinct pathogens, $O_k$, for $k \in [0,n]$, statistical independence corresponds to assuming the infection load of a single individual follows the one-parameter, binomial model $\text{Bin}(n,p)$, in which $p$ is the pathogen prevalence (assumed identical for each pathogen and fitted appropriately to the data), and $n$ is the maximum number of infections that is possible (i.e., the total number of distinct pathogens under consideration). Model predictions are then simply $N$ samples from this binomial distribution, where $N = \Sigma_k O_k$ is the

**Table 2. Sources of data for fitting the NiSP model in which pathogen types, clones, or strains are assumed to be epidemiologically interchangeable.** The data sets include human papillomavirus [22], anther smut (*M. violaceum*) [45], *B. afzelii* on bank voles [46], and malaria (*P. vivax*) [47]. The underlying data for this table can be found in S1 Data.

| Pathogens with *n* distinct types, strains, or clones | *n* | Observed counts, $O_k$ | | | | | | | | | | Total |
| | | 0 | 1 | 2 | 3 | 4 | 5 | 6 | 7 | 8 | 9 | *N* |
|---|---|---|---|---|---|---|---|---|---|---|---|---|
| Human papillomavirus | 25 | 2,933 | 140 | 64 | 26 | 102 | 39 | 12 | 2 | 2 | - | 5,412 |
| Anther smut (*M. violaceum*) | 102 | 285 | 74 | 60 | 32 | 14 | 3 | 3 | 2 | 1 | 1 | 475 |
| *B. afzelii* on bank voles | 7 | 807 | 33 | 26 | 13 | 10 | 11 | 6 | - | - | - | 906 |
| Malaria (*P. vivax*) | 57 | 1,023 | 404 | 291 | 208 | 118 | 50 | 16 | 5 | 1 | 1 | 2,117 |

Abbreviation: NiSP, Noninteracting Similar Pathogens

total number of individuals observed in the data. One interpretation is as a multinomial model in which

$$O_k \sim N q_k \quad \text{where} \quad q_k = C_k^n p^k (1-p)^{n-k}. \tag{28}$$

For the data for malaria corresponding to numbers of individuals, $O_\Gamma$, infected by different sets of pathogens, $\Gamma$, statistical independence corresponds to an *n*-parameter multinomial model, parameterised by the prevalences of the individual pathogens $p_i$ (again fitted to the data), i.e.,

$$O_\Gamma \sim N \prod_{i \in \Gamma} p_i \prod_{i \notin \Gamma} (1 - p_i). \tag{29}$$

**Fitting the models.** The host natural death rate, $\mu$, can be scaled out of the equilibrium prevalences by rescaling time. Fitting the models therefore corresponds to finding value(s) for scaled infection rate(s) $\beta_i$, i.e., $R_{0,i} = \beta_i/\mu$ (all are equal for the NiSP model).

The method used to fit the model does not depend on whether the data are numbers of hosts infected by a particular combination of pathogens or numbers of hosts carrying particular numbers of distinct pathogens, since both can be viewed as *N* samples drawn from a multinomial distribution, with $q_j$ observations of the $j^{th}$ class. If the corresponding probabilities generated by the model being fitted are $p_j$, then the log-likelihood is

$$L = \sum_j q_j \log(p_j). \tag{30}$$

**Table 3. Fitting the NiSP model.** The NiSP model was highly supported over the binomial model ($\Delta AIC \gg 10$) in all cases tested. The final column of the table corresponds to the GoF test of the NiSP model; the value $p > 0.05$ is highlighted in bold and corresponds to lack of evidence for failure to fit the data, and so the NiSP model is adequate for the data concerning human papillomavirus [22].

| | NiSP | | | Binomial | | | GoF |
| | $R_0$ | *L* | *p* | *L* | $\Delta AIC = 2\Delta L$ | *p* | |
|---|---|---|---|---|---|---|---|
| Human papillomavirus | 1.032 | −6,580.9 | 0.031 | −6,868.8 | 575.8 | **0.077** | |
| Anther smut (*M. violaceum*) | 1.008 | −614.0 | 0.008 | −693.3 | 158.6 | 0.001 | |
| *B. afzelii* on bank voles | 1.044 | −652.1 | 0.040 | −799.0 | 293.8 | 0.000 | |
| Malaria (*P. vivax*) | 1.021 | −3,169.2 | 0.021 | −3,467.3 | 596.3 | 0.000 | |

Abbreviations: AIC, Akaike information criterion; NiSP, Noninteracting Similar Pathogens; GoF, goodness of fit

**Table 4. Fitting the NiDP model.** Data sets that are consistent with no interaction between the *Plasmodium* spp. considered are highlighted in grey (and have a row number marked with bold font in the first column). Such data sets have both *p*-values for the GoF test of the NiDP model $p(GoF)>0.05$ (marked in bold in the sixth column), and $\Delta AIC \geq 2$ (for the comparison between the NiDP model and the multinomial model; marked in bold in the 12th column), meaning the NiDP model is adequate. The multinomial model corresponds to the statistical independence hypothesis. Parameters $R_{0,1}$ and $R_{0,2}$ are associated with *P. falciparum* and *P. malariae*, respectively. Parameter $R_{0,3}$ corresponds either to *P. vivax* (upper part of the table, data sets 74–137) or to *P. ovale* (lower part of the table, data sets 68–103). The final column contains a "Y" whenever at least one association between a pair of pathogens was assessed to be significant by Howard and colleagues [17] (and "N" when not significant). A "Y" in cells shaded pink correspond to possible statistical associations that are consistent with our no-interaction model (NiDP), i.e., cases in which our methods lead to results diverging from those reported in [17].

| Row number in Table 1 in [17] | NiDP | | | | | Multinomial | | | | | | Association(s) in [17] |
| --- | --- | --- | --- | --- | --- | --- | --- | --- | --- | --- | --- | --- |
| | $R_{0,1}$ | $R_{0,2}$ | $R_{0,3}$ | $L$ | $p$(GoF) | $p_1$ | $p_2$ | $p_3$ | $L$ | $p$(GoF) | $\Delta$AIC | Y |
| 74 | 1.764 | 1.256 | 1.004 | −340.7 | 0.000 | 0.468 | 0.220 | 0.004 | −311.0 | 0.000 | −59.3 | Y |
| 75 | 1.694 | 1.248 | 1.022 | −194.8 | 0.000 | 0.445 | 0.215 | 0.022 | −177.4 | 0.000 | −34.8 | Y |
| **76** | 1.235 | 1.019 | 1.005 | −492.5 | **0.251** | 0.190 | 0.019 | 0.005 | −493.7 | 0.098 | **2.4** | Y |
| 82 | 1.776 | 1.165 | 1.108 | −996.0 | 0.000 | 0.463 | 0.147 | 0.101 | −936.3 | 0.000 | −119.2 | Y |
| 84 | 1.212 | 1.017 | 1.207 | −684.2 | 0.000 | 0.180 | 0.017 | 0.177 | −660.4 | 0.000 | −47.6 | Y |
| 88 | 1.296 | 1.120 | 1.260 | −314.7 | 0.000 | 0.242 | 0.111 | 0.217 | −295.0 | 0.000 | −39.3 | Y |
| 106 | 1.818 | 1.146 | 1.055 | −4,105.2 | 0.000 | 0.442 | 0.125 | 0.052 | −4,296.6 | 0.000 | 382.9 | Y |
| 108 | 1.241 | 1.024 | 1.096 | −1,147.5 | 0.000 | 0.197 | 0.023 | 0.089 | −1,132.1 | 0.721 | −30.9 | N |
| **109** | 1.023 | 1.013 | 1.045 | −359.3 | **0.866** | 0.023 | 0.013 | 0.043 | −361.1 | 0.343 | **3.5** | Y |
| 111 | 1.198 | 1.005 | 1.786 | −1,929.2 | 0.000 | 0.175 | 0.005 | 0.467 | −1,798.8 | 0.000 | −260.7 | Y |
| **112** | 1.307 | 1.086 | 1.056 | −119.6 | **0.115** | 0.241 | 0.080 | 0.054 | −116.6 | 0.552 | −6.0 | N |
| 113 | 1.213 | 1.007 | 1.119 | −1,324.1 | 0.000 | 0.179 | 0.007 | 0.108 | −1,290.8 | 0.000 | −66.6 | Y |
| 114 | 1.615 | 1.084 | 1.038 | −1,224.4 | 0.000 | 0.392 | 0.080 | 0.037 | −1,182.6 | 0.000 | −83.6 | Y |
| 116 | 1.780 | 1.124 | 1.100 | −1,035.1 | 0.000 | 0.471 | 0.116 | 0.094 | −953.5 | 0.000 | −163.2 | Y |
| 117 | 1.072 | 1.000 | 1.268 | −31,530.5 | 0.000 | 0.068 | 0.000 | 0.214 | −30,958.7 | 0.000 | −1,143.5 | Y |
| **118** | 1.085 | 1.039 | 1.171 | −225.3 | **0.990** | 0.078 | 0.037 | 0.146 | −227.5 | 0.515 | **4.5** | Y |
| 119 | 1.433 | 1.164 | 1.375 | −265.7 | 0.000 | 0.325 | 0.146 | 0.291 | −249.0 | 0.146 | −33.6 | Y |
| 123 | 1.016 | 1.055 | 1.098 | −6,684.7 | 0.000 | 0.016 | 0.052 | 0.090 | −6,623.5 | 0.000 | −122.4 | Y |
| 124 | 1.254 | 1.100 | 1.082 | −3,600.6 | 0.000 | 0.206 | 0.092 | 0.076 | −3,541.3 | 0.017 | −118.7 | Y |
| 127 | 1.341 | 1.005 | 1.266 | −1,087.4 | 0.000 | 0.265 | 0.005 | 0.219 | −1,039.0 | 0.000 | −96.8 | Y |
| **130** | 1.013 | 1.002 | 1.350 | −352.7 | **0.978** | 0.013 | 0.002 | 0.259 | −353.7 | 0.636 | **2.0** | N |
| **132** | 1.397 | 1.027 | 1.074 | −591.8 | **0.347** | 0.285 | 0.026 | 0.068 | −594.3 | 0.067 | **4.9** | Y |
| 133 | 1.571 | 1.022 | 1.332 | −687.9 | 0.000 | 0.375 | 0.022 | 0.257 | −676.2 | 0.001 | −23.4 | Y |
| 137 | 1.196 | 1.005 | 1.130 | −2,356.8 | 0.000 | 0.166 | 0.005 | 0.117 | −2,309.6 | 0.000 | −94.3 | Y |
| **68** | 1.910 | 1.091 | 1.021 | −152.0 | **0.200** | 0.469 | 0.082 | 0.020 | −157.8 | 0.002 | **11.7** | Y |
| **69** | 4.827 | 1.443 | 1.036 | −177.2 | **0.822** | 0.796 | 0.310 | 0.035 | −181.4 | 0.121 | **8.5** | Y |
| **70** | 4.612 | 1.203 | 1.089 | −239.2 | **0.953** | 0.781 | 0.168 | 0.082 | −247.4 | 0.012 | **16.4** | Y |
| 71 | 6.070 | 1.370 | 1.181 | −310.1 | 0.001 | 0.822 | 0.261 | 0.148 | −336.2 | 0.000 | 52.1 | N |
| 77 | 14.275 | 1.383 | 1.142 | −155.3 | 0.032 | 0.944 | 0.286 | 0.127 | −150.4 | 0.931 | −9.9 | N |
| **78** | 4.171 | 1.178 | 1.006 | −166.2 | **0.264** | 0.773 | 0.153 | 0.006 | −163.2 | 0.997 | −5.8 | N |
| **79** | 1.855 | 1.033 | 1.005 | −1,260.1 | **0.969** | 0.461 | 0.032 | 0.005 | −1,263.4 | 0.224 | **6.6** | Y |
| **80** | 1.546 | 1.062 | 1.021 | −715.1 | **0.735** | 0.355 | 0.059 | 0.020 | −715.2 | 0.675 | 0.2 | Y |
| **95** | 1.855 | 1.033 | 1.005 | −1,260.1 | **0.970** | 0.461 | 0.032 | 0.005 | −1,263.4 | 0.224 | **6.6** | N |
| 96 | 1.910 | 1.071 | 1.017 | −240.4 | 0.019 | 0.469 | 0.065 | 0.016 | −248.9 | 0.000 | 17.1 | N |
| **97** | 1.952 | 1.077 | 1.004 | −242.6 | **0.568** | 0.486 | 0.071 | 0.004 | −246.7 | 0.031 | **8.3** | Y |
| **98** | 1.662 | 1.014 | 1.018 | −183.7 | **0.373** | 0.396 | 0.013 | 0.018 | −187.0 | 0.030 | **6.6** | N |
| **99** | 1.627 | 1.019 | 1.019 | −133.7 | **0.823** | 0.384 | 0.019 | 0.019 | −135.6 | 0.332 | **3.8** | N |
| **100** | 1.037 | 1.003 | 1.000 | −432.1 | **0.254** | 0.035 | 0.003 | 0.000 | −433.2 | 0.083 | **2.3** | Y |
| 101 | 3.590 | 1.269 | 1.063 | −11,014.1 | 0.000 | 0.720 | 0.211 | 0.060 | −11,392.7 | 0.000 | 757.3 | Y |
| **102** | 2.473 | 1.153 | 1.027 | −8,188.9 | **0.403** | 0.595 | 0.132 | 0.027 | −8,352.0 | 0.000 | **326.2** | Y |
| 103 | 1.798 | 1.180 | 1.015 | −7,425.7 | 0.000 | 0.437 | 0.150 | 0.015 | −7,736.6 | 0.000 | 621.8 | Y |

Abbreviations: AIC, Akaike information criterion; GoF, goodness of fit; NiDP, Noninteracting Distinct Pathogens.

The models were fitted by maximising $L$ via *optim()* in R [75]. Convergence to a plausible global maximum was checked by repeatedly refitting the model from randomly chosen starting sets of parameters. All models were fitted in a transformed form to allow only biologically meaningful values of parameters; i.e., the basic reproduction numbers were estimated after transformation with $\log(R_{0,i}-1)$ to ensure $R_{0,i}>1$.

**Model comparison.**   To compare the best-fitting NiSP or NiDP model and an appropriate model assuming statistical independence (binomial or multinomial), we use the Akaike information criterion $AIC = 2k - 2\hat{L}$, in which $\hat{L}$ is the log-likelihood of the best-fitting version of each model and $k$ is the number of model parameters. This is necessary because these comparisons involve pairs of models that are not nested.

**Goodness of fit.**   We use a Monte-Carlo technique to estimate *p*-values for model goodness of fit, generating 1,000,000 independent sets of samples of total size $N$ from the multinomial distribution corresponding to the best-fitting model, calculating the likelihood (Eq 30) of each of these synthetic data sets, and recording the proportion with a smaller value of $L$ than the value calculated for the data [76]. This was done using the function *xmonte*() in the R package *XNomial* [77].

## Sources of data and results of model fitting

**Numbers of distinct pathogens (NiSP model).**   Results of fitting the NiSP model to data from four publications for strains of a single pathogen are presented in Fig 3. Error bars are 95% confidence intervals using exact methods for binomial proportions via *binconf()* in the R package *Hmisc* [78]. Results for three further data sets concerning different pathogens of a single host [46, 48, 50] are provided in Text S2 Section 1 (see also S4 Fig).

For convenience, the raw data as extracted for use in model fitting are retabulated in Table 2. Results of model fitting are summarised in Table 3. We used the value $n = 102$ for the number of distinct strains by López-Villavicencio and colleagues [45] following personal communication with the authors; there might be undetected genetic differences due to missing data—which would require a larger value of $n$ in our model-fitting procedure—but we confirmed that our inferences are unaffected by taking any value of $n \in [100,200]$.

**Combinations of pathogens (NiDP model).**   Howard and colleagues [17] report results of analysing 73 data sets concerning multiple *Plasmodium* spp. causing malaria (rows 68–140 of Table 1 in that paper). We reanalysed the subset of these studies satisfying certain additional constraints as detailed in the main text (see S2 Text, Section 2, for a full description of how the studies were filtered). This left a final total of 41 data sets taken from 35 distinct papers: 24 data sets considering the three-way interaction between *P. falciparum*, *P. malariae*, and *P. vivax* and 17 data sets considering the three-way interaction between *P. falciparum*, *P. malariae*, and *P. ovale*.

We used our method based on the NiDP model to test whether any of these data sets were consistent with no interaction between the *Plasmodium* spp. considered (Table 4). We found 15 data sets for which the NiDP model was (1) a better fit than the multinomial model as indicated by ΔAkaike information criterion (AIC) $\geq$ 2 and (2) sufficient to explain the data as revealed by our goodness of fit test. In these 15 cases, our methods therefore support the hypothesis of no interaction. For 11 of these 15 data sets (76, 109, 118, 130, 132, 68, 69, 70, 79, 95, 97, 98, 99, 100, 102), the results as reported by Howard and colleagues [17] instead suggest the strains interact.

## Code availability

Code illustrating all statistical methods is freely available at https://github.com/nikcunniffe/Coinfection.

## Supporting information

**S1 Text. Mathematical supplements.** Further mathematical details on the models considered in the main text, as well as showing how the models can be extended to account for pathogen-specific rates of clearance.
(PDF)

**S2 Text. Sources of data and side results of model fitting.** Gives more details on how data was selected and extracted, as well as discussing additional results of fitting the models that are not presented in the main text.
(PDF)

**S1 Fig. Numerical investigation of the switching time in the deterministic two-pathogen model.** Panels (A) and (C) show how the switching time was calculated for both 'random' (A) and 'one pathogen is invading' (C) initial conditions (described in S1 Text Section 1.2) with epidemiological parameters chosen via a randomisation procedure (which ensured $R_{0,1}$ and $R_{0,2}$ were independently uniformly distributed between 1 and 5). The distribution of switching times over a large number of replicates (B and D) show the switching time is always less than the mean lifetime of an individual host for both initial condition scenarios. In both cases, any transient is therefore likely to have only limited impact (see also S1 Text Section 6).
(TIF)

**S2 Fig. Impact of environmental stochasticity on the deviation between the density of coinfecteds and product of the prevalences in a stochastic two-pathogen model.** The stochastic differential equation version of the two-pathogen model was simulated $10^3$ times, in a population $N = 1,000$, but the individual epidemiological parameters $\beta_1$, $\beta_2$, and $\mu$ were allowed to vary according to the Cox-Ingersoll-Ross process in Eq S77 in S1 Text (with mean values following the parameterisation used in Fig 2 of the main text). The three rows show results for $\sigma = 0$ (i.e., no environmental noise), $\sigma = 0.25$ (i.e., intermediate environmental noise), and $\sigma = 0.5$ (i.e., relatively high environmental noise). (A, E, and I) The evolution of the parameters over time in an individual replicate simulation. (B, F, and J) The corresponding trajectories for the density of infected hosts. (C, G, and K) The distribution of $10^3$ point estimates of $(I_1, I_2)$ when $T = 10$. (D, H, and L) The empirical distribution of the relative deviation from statistical independence $\Lambda = (\bar{J}_{1,2} - \bar{P})/\bar{P}$ over the $10^3$ simulations at each level of noise. For all three levels of noise, the full distributions of $\Lambda$ remain reliably above zero. (Note that since the level of noise is set to zero for the results shown in the top row, panels B, C, and D essentially replicate Fig 2B, 2C and 2D in the main text).
(TIFF)

**S3 Fig. Impact of transient behaviour on the deviation between the density of coinfecteds and product of the prevalences.** The stochastic differential equation version of the two-pathogen model with the parameterisation used in Fig 2 of the main text was simulated 1,000 times with random initial conditions, in a population $N = 1,000$. The 95% interval on the value of $\Lambda$ as extracted from individual simulations at different times is shown for different assumptions on the initial conditions (see also S1 Text, Section 1.2). (A) Random initial conditions, with densities of all four state variables chosen at random. (B) One pathogen is invading the other, which is initially at equilibrium.
(TIF)

**S4 Fig. Comparing the best-fitting NiSP model with a binomial model (i.e., statistical independence) for data sets in which different pathogens are considered.** Model-fitting results are shown for (A) pathogens of *Ixodes ricinus* ticks [50], (B) barley and cereal yellow dwarf

viruses [49], and (C) human respiratory viruses [48]. Insets to each panel show a 'zoomed-in' section of the graph corresponding to high multiplicities of pathogen coinfection, using a logarithmic scale on the y-axis for clarity. Asterisks indicate predicted counts smaller than 0.1. For the data shown in (A), there is no evidence that the NiSP model does not fit the data, and so our test indicates the pathogens do not interact. For the data shown in (B), although the NiSP model is a better fit to the data than the binomial model, there is evidence of lack of goodness of fit, and so our test indicates these pathogens interact (or are epidemiologically different). For the data shown in (C), although the binomial model is a better fit to the data than the NiSP model, there is evidence of lack of goodness of fit, and again it can be concluded that these pathogens interact (or are epidemiologically different). The underlying data for this figure can be found in S9 Data, S10 Data, and S11 Data. NiSP, Noninteracting Similar Pathogens. (TIF)

**S5 Fig. Comparing the best-fitting two-parameter NiSP model with a binomial model (i.e., statistical independence).** Model-fitting results are shown for human respiratory viruses [48]. The inset shows a 'zoomed-in' section of the graph corresponding to high multiplicities of pathogen coinfection, using a logarithmic scale on the y-axis for clarity. The best-fitting NiSP model converges to the binomial model in this case (which is a special case of NiSP for $\mu = 0$, see S1 Text [Section 4]). The underlying data for this figure can be found in S12 Data. NiSP, Noninteracting Similar Pathogens. (TIF)

**S1 Table. Sources of data for fitting the NiSP model in which pathogen species, clones, or strains are assumed to be epidemiologically interchangeable, even though different pathogens are considered.** The data sets include pathogens of *I. ricinus* ticks [50], barley yellow dwarf viruses [49], and respiratory viruses [48]. The underlying data for this table can be found in S1 Data. NiSP, Noninteracting Similar Pathogens. (PDF)

**S2 Table. Additional examples of fitting the NiSP model.** The NiSP model was highly supported over the binomial model ($\Delta AIC \gg 10$) in all cases tested but one (respiratory viruses), in which the binomial model is highly supported over the NiSP model. The final column of the table corresponds to the GoF test of the NiSP model; values $p > 0.05$ correspond to lack of evidence for failure to fit the data, and so the NiSP model is adequate for the data concerning pathogens of *I. ricinus* ticks [50]. AIC, Akaike information criterion; GoF, goodness of fit; NiSP, Noninteracting Similar Pathogens. (PDF)

**S3 Table. Data sets as extracted from the source references for studies focusing on interactions between *P. falciparum*, *P. malariae*, and either *P. vivax* (i.e., 'FMV') or *P. ovale* (i.e., 'FMO').** The asterisks indicate that the corresponding data sets were extracted from the large compendium collated in 1930 by Knowles and White. The number in the leftmost column shows the number of the relevant row in Table 1 of [17]. The rows with (!) correspond to studies for which the total number of individuals sampled as reported by [17] do not match what we found on interrogating the original paper; in all cases, we used the corrected values as shown in the table. The notation 'X' (in FX, MX, or FMX) corresponds either to 'V' (i.e., *P. vivax*, upper part of the table, data sets 74–137) or to 'O' (i.e., *P. ovale*, lower part of the table, data sets 68–103). The underlying data for this table can be found in S2 Data. (PDF)

**S4 Table. Fitting the NiSP model to data sets corresponding to human papillomavirus [22], pathogens of *I. ricinus* ticks [50], anther smut (*M. violaceum*) [45], barley yellow dwarf viruses [49], *B. afzelii* on bank voles [46], malaria (*P. vivax*) [47], and human respiratory viruses [48].** Parameters for the best-fitting variant of the NiSP model for each pathogen species, strain, or clone are highlighted in bold; the two-parameter model is supported in cases for which $p<0.05$ in the 'Model Selection' part of the table (including human papillomavirus and malaria [*P. vivax*]). The NiSP model was highly supported over the binomial model ($\Delta AIC \gg 10$) in all cases tested but one (human respiratory viruses). The final column of the table corresponds to the GoF test of the best-fitting model; values $p>0.05$ correspond to lack of evidence for failure to fit the data, and so the NiSP model is adequate for the data concerning human papillomavirus and pathogens of *I. ricinus* ticks. These results are qualitatively identical to those for the model without specific clearance as presented in the main text. Note that in the NiSP model, $\beta$ and $\gamma$ are scaled relative to $\mu$. This is why $\beta$ and $\gamma$ of NiDP reach extremely high values for respiratory viruses. Parameter estimation tends to $\mu = 0$, which actually corresponds to the binomial model, which has one fewer parameter (see S1 Text Section 4.5 and S5 Fig). Hence, $\Delta AIC = -2$ for respiratory viruses, since the NiSP model requires one additional parameter compared to the binomial model. AIC, Akaike information criterion; GoF, goodness of fit; NiDP, Noninteracting Distinct Pathogens; NiSP, Noninteracting Similar Pathogens. (PDF)

**S1 Data. Data underlying Table 2 and S1 Table.** (CSV)

**S2 Data. Data underlying S3 Table.** (CSV)

**S3 Data. Data underlying Fig 3A.** (CSV)

**S4 Data. Data underlying Fig 3B.** (CSV)

**S5 Data. Data underlying Fig 3C.** (CSV)

**S6 Data. Data underlying Fig 3D.** (CSV)

**S7 Data. Data underlying Fig 4A.** (CSV)

**S8 Data. Data underlying Fig 4B.** (CSV)

**S9 Data. Data underlying S4A Fig.** (CSV)

**S10 Data. Data underlying S4B Fig.** (CSV)

**S11 Data. Data underlying S4C Fig.** (CSV)

**S12 Data. Data underlying S5A Fig.** (CSV)

## Acknowledgments

We thank S. Alizon, T. Berrett, E. Bussell, V. Calcagno, R. Donnelly, T. Giraud, M. López-Villavicencio, T. Obadia, M. Parry, M. Plantegenest, O. Restif, E. Seabloom, J. Shykoff, R. Thompson, and C. Trotter for helpful discussions or provision of data.

## Author Contributions

**Conceptualization:** Frédéric M. Hamelin, Linda J. S. Allen, Vrushali A. Bokil, Louis J. Gross, Frank M. Hilker, Michael J. Jeger, Carrie A. Manore, Alison G. Power, Megan A. Rúa, Nik J. Cunniffe.

**Formal analysis:** Frédéric M. Hamelin, Linda J. S. Allen, Nik J. Cunniffe.

**Investigation:** Frédéric M. Hamelin, Linda J. S. Allen, Nik J. Cunniffe.

**Methodology:** Frédéric M. Hamelin, Linda J. S. Allen, Nik J. Cunniffe.

**Software:** Nik J. Cunniffe.

**Visualization:** Frédéric M. Hamelin, Nik J. Cunniffe.

**Writing – original draft:** Frédéric M. Hamelin, Linda J. S. Allen, Nik J. Cunniffe.

**Writing – review & editing:** Frédéric M. Hamelin, Linda J. S. Allen, Vrushali A. Bokil, Louis J. Gross, Frank M. Hilker, Michael J. Jeger, Carrie A. Manore, Alison G. Power, Megan A. Rúa, Nik J. Cunniffe.

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
