## [Editor Report · Decision Letter 0]

23 Aug 2019

Dear Dr Cunniffe, 

Thank you for submitting your manuscript entitled "Co-infections by non-interacting pathogens are not independent and require new tests of interaction" for consideration as a Research Article by PLOS Biology.

Your manuscript has now been evaluated by the PLOS Biology editorial staff as well as by an academic editor with relevant expertise and I am writing to let you know that we would like to send your submission out for external peer review.

*Please be aware that, due to the voluntary nature of our reviewers and academic editors, manuscripts may be subject to delays during the holiday season. Thank you for your patience.*

Please re-submit your manuscript within two working days, i.e. by Aug 25 2019 11:59PM.

Kind regards,

Lauren A Richardson, Ph.D

Senior Editor

PLOS Biology

---

## [Decision Letter · Decision Letter 1]

23 Sep 2019

Dear Dr Cunniffe,

Thank you very much for submitting your manuscript "Co-infections by non-interacting pathogens are not independent and require new tests of interaction" for consideration as a Research Article at PLOS Biology. Your manuscript has been evaluated by the PLOS Biology editors, an Academic Editor with relevant expertise, and by three independent reviewers.

In light of the reviews (below), we are pleased to offer you the opportunity to address the comments from the reviewers in a revised version that we anticipate should not take you very long. Some additional analysis and plots may be necessary to fully address reviewer 2 and 3's comments. We will then assess your revised manuscript and your response to the reviewers' comments and we may consult the reviewers again.

Your revisions should address the specific points made by each reviewer. Please submit a file detailing your responses to the editorial requests and a point-by-point response to all of the reviewers' comments that indicates the changes you have made to the manuscript. In addition to a clean copy of the manuscript, please upload a 'track-changes' version of your manuscript that specifies the edits made. This should be uploaded as a "Related" file type. You should also cite any additional relevant literature that has been published since the original submission and mention any additional citations in your response. 

Before you revise your manuscript, please review the following PLOS policy and formatting requirements checklist PDF: http://journals.plos.org/plosbiology/s/file?id=9411/plos-biology-formatting-checklist.pdf. It is helpful if you format your revision according to our requirements - should your paper subsequently be accepted, this will save time at the acceptance stage.

Please note that as a condition of publication PLOS' data policy (http://journals.plos.org/plosbiology/s/data-availability) requires that you make available all data used to draw the conclusions arrived at in your manuscript. If you have not already done so, you must include any data used in your manuscript either in appropriate repositories, within the body of the manuscript, or as supporting information (N.B. this includes any numerical values that were used to generate graphs, histograms etc.). For an example see here: http://www.plosbiology.org/article/info%3Adoi%2F10.1371%2Fjournal.pbio.1001908#s5.

For manuscripts submitted on or after 1st July 2019, we require the original, uncropped and minimally adjusted images supporting all blot and gel results reported in an article's figures or Supporting Information files. We will require these files before a manuscript can be accepted so please prepare them now, if you have not already uploaded them. Please carefully read our guidelines for how to prepare and upload this data: https://journals.plos.org/plosbiology/s/figures#loc-blot-and-gel-reporting-requirements.

Upon resubmission, the editors assess your revision and assuming the editors and Academic Editor feel that the revised manuscript remains appropriate for the journal, we may send the manuscript for re-review. We aim to consult the same Academic Editor and reviewers for revised manuscripts but may consult others if needed.

We expect to receive your revised manuscript within one month. Please email us (plosbiology@plos.org) to discuss this if you have any questions or concerns, or would like to request an extension. At this stage, your manuscript remains formally under active consideration at our journal; please notify us by email if you do not wish to submit a revision and instead wish to pursue publication elsewhere, so that we may end consideration of the manuscript at PLOS Biology.

When you are ready to submit a revised version of your manuscript, please go to https://www.editorialmanager.com/pbiology/ and log in as an Author. Click the link labelled 'Submissions Needing Revision' where you will find your submission record. 

Sincerely,

Di Jiang, PhD

Associate Editor

on behalf of 

Lauren A Richardson, Ph.D

Senior Editor

PLOS Biology

Reviewer remarks:

Reviewer #1: This manuscript tackles an issue of growing recognition - how to infer the presence of interspecific parasite interactions from cross-sectional data. This is a problem since, particularly for wildlife systems, longitudinal or experimental data are rare, so we often have to rely on data of this form to draw conclusions of how and to what extent co-infecting parasites interact with each other. This ms very neatly demonstrates that the fundamental assumption that non-interacting parasites should show statistical independence in their co-occurrence is wrong, due to the likely positive correlations that arise due to the confounding effect of host age. The authors then show how this non statistical independence can be accounted for, nicely illustrated with re-fitting models to previously published data. 

Overall this is a very well written, elegant and informative ms, and I think it very clearly makes important points about some of our fundamental assumptions about our null models regarding patterns of co-association between non-interacting species. I only have a couple of very minor additional suggestions:

- Equation 4 is presented as the measure of deviation of the prevalence of co-infection from that of statistical independence, and how this is affected by host longevity - that's fine, but I found Equation 9 (in the Methods) to be a much nicer/clearer representation, clearly showing that if mu=0 (essentially immortal hosts) then the observed co-infection prevalence equals the expected proportion assuming independence - whereas as mu increases (mean host lifespan decreases) this difference accentuates. I wonder if Eq 9 can be brought into the main text to help reinforce these points.

- Line 153 states the "deviation [between co-infection prevalence and that required by statistical independence] is zero if an only if the host natural death rate = 0". Yes, that's strictly true, but (from Eqns 4/9] it would also be approximately true if the R0 and/or transmission rates of either or both pathogens are large - the deviation tends to zero as beta1 and/or beta2 increase. Biologically this means we would expect to see the assumption of statistical independence violated for pathogens with high transmission rates/R0s, whereas those with very low transmission rates (assuming R0 remains > 1) we would likely not see much departure from the assumption of statistical independence. It might be useful to make this point explicitly here.

- I found it very interesting that this ms highlights the expectation of positive correlations among parasites, even if they don't interact (or presumably even if they negatively interact, providing the confounding age effect is sufficiently strong) - as this matches the finding in Fenton et al 2014 IJP that most correlation-based cross-sectional analyses revealed positive associations between pairs of parasites that are known to interact negatively with each other (their Fig 2). I wonder if it's useful to make this point in the Discussion somewhere, as a theoretical explanation for this empirically-observed result.

Reviewer #2: This paper uses SIS models to show why systems of non-interacting pathogens will—counter to intuition—always appear to exhibit statistical dependence according to conventional metrics, simply on account of positive correlations between I1 and I2 that are generated when coinfected hosts die. The authors go on to develop some statistical tests based on this insight that can account for this effect, and they re-analyze a number of datasets with the new tests. The problem is tackled quite thoroughly by the authors, who present several lines of methodological evidence along with numerical simulations and empirical analyses. I think this paper will be of broad interest to the infectious disease modelling community, and even more broadly in population biology and epidemiology. The paper is also very well-written and the figures are clear. I only have a few minor revisions to suggest: 

1. The analysis relies heavily on the SIS natural history which is suitable for chronic infections (as the authors point out). However, they apply their test to HPV, which exhibits short-lived natural immunity followed by clearance, in most cases. In the Discussion section, the authors should discuss in more detail how other natural histories would influence their findings, including their finding of independence for HPV strains. I assume it would make their findings conservative. 

2. The proof in S1.1 only shows that Z becomes negative after a finite time t1, meaning that it could be positive before that. Hence, depending on how large t1 is, we could observe either J1,2 > I1*I2 or vice versa. Please comment and explain whether this is a significant limitation, and adjust writing in main text as needed. 

3. Following on comment #2, the authors could either expand their numerical analysis depicted in Figure 2 to explore whether transient dynamics have any interesting impact on the relationship between J1,2 and I1*I2. Alternatively, they could include some speculation about the impact of transients in the Discussion section.

Reviewer #3: It was a pleasure to read the manuscript by Hamelin et al. touching on the important question in ecology and epidemiology whether co-infections of non-interacting pathogens are statistically independent. Building upon prior work by Kucharski et al., the manuscript showcases a wide range of tools from ODE and SDE modelling to statistics. It provides a practical tool to test for interactions between pathogens based on cross-sectional data for chronic infectious diseases explicitly accounting for non-independence of co-infection prevalences. The availability of source code and data gathered from previous studies will further benefit the research community. 

Major comments

In the SDE model, especially Figure 2B, the authors show only a single replicate to underpin non-independence, i.e. that the product prevalence is lower the the co-infection prevalence. It would be interesting to see a more representative number of replicates (with mean and confidence interval bands). Would then the product prevalence still be statistically significantly different from the the co-infection prevalence? 

Concerning the positive correlation between pathogen prevalences resulting from the covariance matrix of fluctuations near the equilibrium, the authors mentioned the works of O’Dea et al. (2018). In the cited paper, assumptions on environmental (multiplicative gamma white) noise play an import role in the calculation of the covariance matrix. Would you expect that the absence of biological interactions between pathogens (as seen in the HPV example) would still be identifiable with your approach if your model was augmented by such environmental factors? 

Similarly, in your discussion section you touch upon whether structural model constraints (e.g. age classes) should necessarily be accounted for in order to test for non-interaction between pathogens. It would be interesting to sketch strategies how to deal with structures (e.g. networks, meta-populations, immunity) that are relevant for chronic diseases and that might mask interactions.

Minor comments

Main text

line 343 replace “of of” by “of”

Supplementary materials

S.1.4.3

In equation (S37) the first term on the right-hand side lacks a minus sign. The same holds for the following equalities, the first terms lack a minus sign.

S.1.4.4

It is not clear whether this paragraph concerns the case of negligible mortality, as μ does not appear in equation (S38). On the other hand, you refer later in line 241 to the specific case of μ=0.

---

## [Editor Report · Decision Letter 2]

18 Oct 2019

Dear Dr Cunniffe,

Thank you for submitting your revised Research Article entitled "Co-infections by non-interacting pathogens are not independent and require new tests of interaction" for publication in PLOS Biology. 

The Academic Editor and I have now assessed your revision, and we are delighted to let you know that we are essentially satisfied and will not require re-review. We do ask that you 1) explicitly mention the focus on chronic infections in the abstract and 2) on l290 ('we note our analysis indicates a high-profile model of May and Nowak'), clarify what 'high-profile' means here (e.g. 'commonly used' or 'oft-cited'). We will publish your study, assuming you are willing to make these final edits and those needed to meet our production requirements. Congratulations!

Before we can formally accept your paper and consider it "in press", we also need to ensure that your article conforms to our guidelines. A member of our team will be in touch shortly with a set of requests. As we can't proceed until these requirements are met, your swift response will help prevent delays to publication.

Please note that you may have the opportunity to make the peer review history publicly available. The record will include editor decision letters (with reviews) and your responses to reviewer comments. If eligible, we will contact you to opt in or out.

Sincerely,

Lauren A Richardson, Ph.D

Senior Editor

PLOS Biology

DATA POLICY:

Regardless of the method selected, please ensure that you provide the individual numerical values that underlie the summary data displayed in the following figure panels: (e.g. Figs. 3A-D, 4AB, S4A-C, S5), as they are essential for readers to assess your analysis and to reproduce it. 

Please also ensure that figure legends in your manuscript include information on where the underlying data can be found.

Please ensure that your Data Statement in the submission system accurately describes where your data can be found and includes a list of all the data sources.

---

## [Editor Report · Decision Letter 3]

4 Nov 2019

Dear Dr Cunniffe,

On behalf of my colleagues and the Academic Editor, Adam J Kucharski, I am pleased to inform you that we will be delighted to publish your Research Article in PLOS Biology. 

Early Version

PRESS 

Kind regards,

Sofia Vickers

Senior Publications Assistant

PLOS Biology

On behalf of, 

Lauren Richardson,

Senior Editor

PLOS Biology